# Social Isolation and Social Support Influence Health Service Utilisation and Survival after a Cardiovascular Disease Event: A Systematic Review

**DOI:** 10.3390/ijerph20064853

**Published:** 2023-03-09

**Authors:** Rosanne Freak-Poli, Jessie Hu, Aung Zaw Zaw Phyo, S. Fiona Barker

**Affiliations:** 1School of Clinical Sciences at Monash Health, Monash University, Melbourne, VIC 3168, Australia; 2Department of Epidemiology and Preventive Medicine, School of Public Health and Preventive Medicine, Monash University, Melbourne, VIC 3004, Australia

**Keywords:** social isolation, social support, loneliness, cardiac rehabilitation, cardiovascular diseases, rehabilitation, interpersonal relations

## Abstract

Both cardiovascular disease (CVD) and social health carry high health and economic burdens. We undertook a systematic review to investigate the association between social isolation, low social support, and loneliness with health service utilisation and survival after a CVD event among people living in Australia and New Zealand. Four electronic databases were systematically searched for the period before June 2020. Two reviewers undertook the title/abstract screen. One reviewer undertook a full-text screen and data extraction. A second author checked data extraction. Of 756 records, 25 papers met our inclusion criteria. Included studies recruited 10–12,821 participants, aged 18–98 years, and the majority were males. Greater social support was consistently associated with better outcomes on four of the five themes (discharge destination, outpatient rehabilitation attendance, rehospitalisation and survival outcomes; no papers assessed the length of inpatient stay). Positive social health was consistently associated with better discharge designation to higher independent living. As partner status and living status did not align with social isolation and social support findings in this review, we recommend they not be used as social health proxies. Our systematic review demonstrates that social health is considered in cardiac care decisions and plays a role in how healthcare is being delivered (i.e., outpatient, rehabilitation, or nursing home). This likely contributes to our finding that lower social support is associated with high-intensity healthcare services, lower outpatient rehabilitation attendance, greater rehospitalisation and poorer survival. Given our evidence, the first step to improve cardiac outcomes is acknowledging that social health is part of the decision-making process. Incorporating a formal assessment of social support into healthcare management plans will likely improve cardiac outcomes and survival. Further research is required to assess if support person/s need to engage in the risk reduction behaviours themselves for outpatient rehabilitation to be effective. Further synthesis of the impact of social isolation and loneliness on health service utilisation and survival after a CVD event is required.

## 1. Introduction

Cardiovascular disease (CVD) is the leading cause of death globally [1] and carries a high economic burden, which is driven largely by productivity and direct healthcare costs [2,3]. It is essential to identify factors that influence health service utilisation and subsequent CVD events, to mitigate the significant health and economic burden of CVD. Poor social health is a modifiable risk factor for CVD [4] and mortality [5] and also carries a high health and economic burden. Social health refers to a person’s ability to form fulfilling, meaningful relationships; a person’s ability to adapt in social situations; and support from other people, institutions, and services. The concepts of social isolation, loneliness, and social support are often discussed in relation to social health. How these concepts are defined can be beneficial for cardiovascular disease patients [6]. Social isolation can be defined as an objective measure of the number or quantity of social connections [7,8]. Social support can be defined as a subjective measure of how social connections are operationalised, while loneliness can be defined as a subjective negative experience related to isolation or lack of social support [7,8]. There is a depth of research describing different facets of these social health concepts. Social isolation has been described as having a perceptive component based on experience over the life course, for example, family role-modelling socialising behaviours, family dynamics, and long-term patterns of socialisation [9]. The perceptive component then impacts the objective measurement of the number of social contacts or social relationships [9]. Hence, the social isolation could be argued to also be subjective based on personal history and expectations. Similarly, loneliness has been described as having two components. Social loneliness is related to a subjective lack of social isolation—“*the quantitative aspect of loneliness, that is, the subjective lack of a broader support network*” [10]. Emotional loneliness is related to a subjective lack of social support—“*described as a subjective feeling of detachment, thereby representing a qualitative aspect of loneliness pertaining to the lack of deep and meaningful relationships*” [7,10]. Further, social support is not always a positive experience and can be separated into positive and negative support [11]. Family dynamics, personal expectations and communicative norms, for example, can have negative emotional, behavioural and cognitive consequences. People can feel that the support is unhelpful or creates social constraints [11]. Despite the ample literature describing these social health concepts, they have been historically assessed as substitutes in health research; however, research has emerged demonstrating that these concepts should be separated, with different implications for health and well-being [6,7,8,12,13,14,15,16]. Thus, it may be important to focus on one disease outcome. The change in our understanding of these social health constructs as individual, yet interrelated, concepts has also been observed in theoretical frameworks. The Berkman et al. [17] model discusses the individual- and societal-level predisposing factors that contribute to social health constructs jointly, while the more contemporary de Jong Gierveld et al. [18] separates social isolation and loneliness when reviewing the empirical evidence from psychology, sociology and epidemiology. Instruments measuring these social health concepts have been compared. One particularly relevant conceptual framework compares structural or functional aspects and the degree of subjectivity asked of respondents between social health measurement instruments [19].

Generally, people with poor social health use health services more frequently. A systematic review conducted in 2018, from 126 eligible studies, observed an association between older adults’ lower social support and increased rates of readmission to hospital, and between smaller social networks and longer hospital stays [20]. However, the review also observed that social support does not influence ambulatory care (including physician visits and community- or home-based services), beyond what is required based on their health status [20]. Additional themes explored in that review were hospital admissions, emergency department use, contact with general health services, and mental health service use, which were not associated with social support among older adults [20]. A subsequent systematic review in 2018 of 13 studies reported that living alone, social isolation and low social support were each associated with more hospital readmissions among heart failure patients [21]. A systematic review conducted in 2019 identified two (of four) studies that demonstrated low social support as a factor relating to emergency department readmission among older people [22]. A recent systematic review conducted in 2020 reported emerging evidence that loneliness was associated with emergency department use and CVD-specific hospitalization [23]. Another recent systematic review conducted in 2020 reported that greater social health was consistently associated with better mental health outcomes (lower depressive symptoms, anxiety symptoms, and psychological distress) among cardiac patients, but it did not assess health service utilisation [12].

To date, relevant systematic reviews have been focused on social support only [20,22], older adults (aged 60+ [20] or 65+ [22] years), or one specific health service outcome [21,22]. In addition, different types of ill-health and chronic diseases (e.g., cardiovascular disease (CVD), cancer, dementia) can have a different set of social needs [11]. For example, a person’s involvement in the health care process may be influenced by the disease type, as well as the person’s prior experience and expectations of their involvement [11]. Notably, one study found that increasing levels of loneliness predicted an increased confounder-adjusted risk of being hospitalized for CVD, but not for falls or respiratory disease, over a median of 9 years [24].

We aimed to assess whether social isolation, low social support or loneliness are associated with health service utilisation and survival after a CVD event.

## 2. Materials and Methods

The initial aim of the systematic review was to assess the influence of social health on a CVD patient’s journey. The initial searches supplied >1 million potential papers. Following the preferred methodology to reduce the threat to validity from bias and confounding [25], we restricted based on country [12]. Australia and New Zealand were chosen due to the health care system (both free universal/public and private) that is common and reciprocated in many countries [12].

The second iteration of the search resulted in >10 outcomes of interest, as we had not predefined these. To ensure adequate discussion of each outcome, we split the outcomes over two reviews. The other review [12] focuses on the influence of social health on a CVD patient’s health, including physical health, mental health and quality of life. The main finding was that greater social health was consistently associated with better mental health outcomes (lower depressive symptoms, anxiety symptoms and psychological distress) among cardiac patients; the review did not assess health service utilisation [12]. Hence, the methodology has been detailed elsewhere [12] and was conducted in accordance with the registered protocol (Prospero CRD42020099557). The study selection was oriented according to the Preferred Reporting Items for Systematic Reviews and Meta-Analyses Extension for Scoping Reviews (PRISMA) diagram [26]. The only other change since the protocol is that we excluded review articles and included loneliness as an exposure. In brief,

(1)Population: people living in Australia and New Zealand who had experienced a CVD event. Exclusions included intervention studies and cohorts of people with aphasia, as aphasia is not exclusively caused by a CVD event.(2)Exposure: Social health was assessed as a predictor, effect modifier, or mediator of a CVD patient’s health service utilisation or survival.(3)Outcome: CVD-related health services or survival.(4)Search: Four electronic databases from the earliest record to 21 June 2020 Appendix A). References of included papers and relevant review articles were scanned for eligible papers.(5)Screening: Two independent reviewers for titles/abstracts, with discrepancies included. One reviewer for full text.(6)Data extraction: One author (R.F.P) extracted data using a pre-specified form, with a second author either independently extracting or checking data.(7)Authors were contacted when (a) the article was a conference abstract, (b) a full text could not be obtained from two university library sources, or (c) clarification or additional data were required for included papers. Authors were contacted at least twice (via email, LinkedIn, or ResearchGate).(8)Interpretation: During data extraction, papers were grouped based on their outcome/s.

## 3. Results

### 3.1. Results of the Search

Our database search identified 1131 records (724 unique records after removing duplicates; Figure 1). Of these, we considered 320 as potentially eligible based on their title and abstract, which were assessed in a full-text review. The inter-rater reliability was 0.73 (range 0.67–1.00) [27]. Seventeen potentially relevant conference abstracts were identified, and authors were contacted for further details to enable assessment of eligibility, of which eight responded with additional information [28,29,30,31,32,33,34,35,36]. Nine remain unclassified (see Appendix A, for papers awaiting classification). We excluded 284 at full-text review (see Appendix A for excluded papers), and 19 papers met our inclusion criteria. Scanning references of the included papers identified 26 additional papers for full-text review, of which four [37,38,39,40] met our inclusion criteria. Assessment of references from the final included papers did not reveal further potential papers.

After data extraction, we contacted the authors of 11 papers for further clarification through email or LinkedIn. Four responses [41,42,43,44] were received regarding six papers, with authors providing a further seven unique papers to assess, of which two [45,46] met our inclusion criteria.

### 3.2. Included Publications and Study Cohorts

We included 25 publications from 24 study cohorts: 19 from the original search, 4 from references and 2 from contact with authors (Table 1). The included articles were published between 1982 [47] and 2017 [48,49], with 48% published within ten years of the search [48,49,50,51,52,53,54,55,56,57,58].

*Eligibility and recruitment:* Nineteen studies recruited CVD patients [37,38,39,45,46,47,48,49,50,51,52,56,57,58,59,60,61,62,63], nine recruited healthcare workers [40,49,52,53,54,55,59,63,64,65] and one recruited families of CVD patients [49] (three studies recruited a combination of these [49,52,63]). CVD patients were recruited through hospitals (n = 10 [46,47,49,51,56,57,59,60,61,62]), rehabilitation (n = 7 [38,39,45,50,52,58,63]), or alternative locations (n = 2 not described [37,48]). Five study samples were part of cohort studies [37,38,48,56,61].

*CVD measure:* Recruitment was focused on stroke (n = 1 [37,40,45,46,51,52,53,54,55,63,65]), myocardial infarction (n = 6 [38,47,56,59,61,62]; one [38] combined with angina, another combined with coronary artery bypass grafting and percutaneous transluminal coronary angioplasty), coronary disease (n = 2 [58,64]), heart failure (n = 2 [48,57]), percutaneous coronary intervention (n = 1 [60]) or CVD more broadly (n = 3 [39,49,50]).

*Social health measurement:* As several studies referred to partner status or living situation as proxies or measures of social health, we re-read included papers and undertook additional data extraction if partner status or living situation was assessed as an exposure. Studies reported social support (n = 13 [38,40,45,46,47,48,52,54,55,56,58,59,61,65], sometimes labelled ”social situation, including personal and community supports”, plus n = 4 emerged from qualitative [49,53,62,64]), partner status (n = 10 [45,48,50,57,58,59,60,61,62,63], sometimes labelled ”marital status”, plus n = 2 emerged from qualitative [39,64]), living situation (n = 10 [45,50,51,58,59,60,62,65], sometimes labelled ”premorbid living arrangements”), social isolation (n = 4 [46,48,61,66], sometimes labelled ”social interaction” or ”social dysfunction”), next of kin (n = 1 [57]), and loneliness (n = 1 [49]) as exposures or qualitative themes potentially influencing health service utilisation or survival [39,49,52,53,62,64].

One study used the social health scale Perceived Social Support Scale (PSSS) [56]. Studies used social health sub-scales from the Duke Social Support Index (DSSI) [48], Multi-dimensional Outcomes Expectations for Exercise Scale (MOEES) [58], Cardiac Rehabilitation Barriers Scale (CRBS) [58], General Health Questionnaire [61], or the Social Environment Questionnaire (SEQ) [47].

*Demographics:* The majority of studies recruited participants from Australia (n = 21), while only two [37,58] recruited from New Zealand and one [61] recruited from both countries. Studies recruited between 10 [62] and 12,821 [38] participants. Five studies did not report gender [40,49,53,61,64]. Among the studies that reported gender, males were generally more likely to be recruited: one study focused on males [47], seven recruited more males (>60%) [38,50,56,58,59,60,62], 11 recruited roughly evenly by gender [37,39,45,46,48,51,52,54,55,57,63], and one recruited more females (males < 40%) [65]. Fourteen studies [37,39,45,46,48,50,51,52,54,55,56,58,60,61,63] reported their eligibility or sample age range, which ranged between 18 [60] and 98 [51] years. The mean or median age was reported for the patient sample in 15 studies and varied between 46 [54,55] and 82 [48] years.

*Baseline and Duration:* A mix of quantitative (n = 18 [37,38,40,45,46,47,48,50,51,54,55,56,57,58,59,60,61,63,65]) and qualitative (n = 5 [39,49,52,62,64]) studies were included, with one study [52] using mixed methods. The majority of studies were cross-sectional (all of the qualitative, 10 of the [40,45,46,50,51,54,55,58,59,60,63,65]). Longitudinal [37,38,47,48,50,56,57,61] follow-up ranged from 30 days [48] to 21 years [37], with two longitudinal studies [38,61] not reporting the length of follow-up.
ijerph-20-04853-t001_Table 1Table 1Characteristics and relevant findings of included papers.Study IDSampleEligibility and DemographicsSocial Health *OutcomeRelevant FindingsINPATIENT LENGTH OF STAY 


Turner 2010 [50]Baseline July 2003–Jan 2006; Quant; L (6 w, 2.6 y); Au, NSW, Newcastle; n = 293–322 depending on missing dataCardiac; rehabilitation outpatients 27.5%F; age 64.19 + 10.91 yMarital status (n = 293; married, single, widowed, divorced or separated). Living alone (n = 322; yes or no)Hospital length of stay, mean ~8dBeing married was associated to hospital length of stay in univariable analysis (comparator not clear, z = 2.27, *p* = 0.02) for all hospital stays during 2.6 y + 0.9SD follow-up but not after adjustment. No association was observed between living situation and hospital length of stay (statistics not reported).Unsworth 1996 [63]Baseline 1992; Quant; CS; Au, VIC; n = 62 (patients and consultants)Stroke; new cerebrovascular, inpatient rehabilitation > 6 d, age ≥ 60 y; and a patient’s rehabilitation team memberPatients: 52%F; age 75 y (60–90); Team: NRMarital status (married, Single)Rehabilitation length of stay, mean 57 dRehabilitation length of stay was not associated with marital status (t(58) 0.08, *p* = 0.9343). DISCHARGE DESTINATION 


Hakkennes 2013 [52]Baseline June 2010–Sep 2011; Quant & Qual; CS; Au, VIC; n = 89 (75 patients, 14 consultants)Acute severe stroke; admitted (primary diagnosis), onset ≤ 3 d prior to admission, not from high-level residential care, not in intensive/palliative care on day 3 post-stroke. Assessors were clinicians responsible for assessing the suitability of patients for inpatient rehabilitation (43% consultants, 36% registrars, 21% geriatricians). No restrictions regarding qualifications or experiencePatients: 49.3%F; median age 76.5 y IQR 66.0–83.0Living arrangement (alone, with others, supported accommodation). Qual (Social Attribute factor: Pre-morbid living situation, Patient/carer goals, Social support, Patient/carer advocating for rehabilitation)Discharge to inpatient rehabilitation vs. notThose accepted for rehabilitation enrolment vs. not were more likely to be living at home with others pre-stroke (Of accepted: 83.6% Home alone vs. 13.1%, Supported accommodation 3.3%, *p* = 0.041). (Qual) In factor analysis, Social Attributes (4 variables) accounted for 14% of the variance in allocation. An increase by 1 unit on the scale of the factor representing social attributes increased the odds of being discharged to rehabilitation by 4.402 (95%CI:1.436–13.494, *p* = 0.010). For those not accepted for inpatient rehabilitation, social support (median 8.2, IQR 6.7–8.5) was one of three most important items. Hayward 2014 [54,55]Baseline 2012; Quant; CS; Au, QLD; n = 21 consultantsStroke; consultant medical officers45.5%F; age 46.4 + 10.1 yFactors related to social and support networks: Capacity to adapt residence for discharge, Premorbid place of residence (high-level residential care vs. other), Presence of a spouse/carer/relative and: readiness to support, ability to support, expectations for recovery, acceptance of functional prognosis/goal of rehabilitationPerceptions of influences on discharge decision to inpatient rehabilitation vs. notThe social factors favouring the decision to admit a person with stroke to inpatient rehabilitation were capacity to adapt residence for discharge (40.9% favours, 40.9% strongly favours; total = 81.8%); presence of a spouse/carer/relative (40.9% favours, 36.4% strongly favours; total = 77.3%); spouse/carer/relative who was able to provide support (50% favours, 13.6% strongly favours; total = 63.6%); and spouse/carer/relative’s acceptance of the functional prognosis/goal of rehabilitation (50% favours, 22.7% strongly favours; total = 72.7%). The social factors disfavouring the decision to admit a person with stroke to inpatient rehabilitation were the spouse/carer/relative being ready to support (9.1% strongly disfavours, 63.6% disfavours; total = 72.7%) and have high/unrealistic expectations for recovery (68.2% strongly disfavours, 22.7 disfavours; total = 90.9%).Ilett 2010 [51] Baseline NR; Quant; CS; Au, VIC; n = 616Stoke (primary diagnosis); hospitalised, survivors, onset < 3 d, symptoms not resolved by day 3, not from residential care or another hospital, not admitted with another primary illness or incident47%F; age 72.2 + 12.7 y (22–98)Living circumstance (alone, family, supported accommodation/hostel)Discharge destination from hospital (home, rehabilitation, nursing home)Patients who lived with family pre-stroke were more likely to be discharged to rehabilitation than a nursing home compared to patients who lived in supported accommodation/hostel (B −1.19, *p* = 0.03; no association between living home alone compared to living in supported accommodation/hostel B −0.93, *p* = 0.11). No association was observed between pre-stroke living situation and discharge to home versus rehabilitation (with family −0.17, *p* = 0.79; alone −0.94, *p* = 0.15, compared to supported accommodation/hostel). Kennedy 2012 [53]Baseline NR; Qual; CS; Au, VIC; n = 17 physicians Stroke; rehabilitation unit physiciansDemographics NRQual (theme: Level of social support)Rehabilitation admission vs. notSocial support was the second most influential patient-based factor (behind prognosis) influencing selection for rehabilitation (rating: median 2nd place, IQR: 1–3)McBride 2017 [49]Baseline July 2010–June 2015; Qual; CS; Au, SA; n = NRCardiac; Aboriginal patients, families, hospital staffDemographics NRQual (theme: Family relationships: support and loneliness)Hospital self-discharge “Potential reasons for [high rates of] self-discharge [by Aboriginal patients] include: competing family and community obligations; a lack of communication on the importance of staying in hospital; grief, loss and fear; loneliness and dislocation from family and community, and; perceptions of inadequate or racist treatment. Active involvement of family, community and Aboriginal staff were key in reversing patient self-discharge.”Unsworth 1993 [40]Baseline 1991; Quant; CS; Au, VIC; n = 82 consultantsStroke; clinicians currently involved in making accommodation decisions or who had been involved in the past two years: 9 rehabilitation physicians, 16 nurses, 9 speech therapists, 19 occupational therapists, 14 physiotherapists and 15 social workers.Demographics not collected, hence NR3 of 15 cues: Premorbid living arrangements; Social situation, including personal and community supports; Relatives’ choice/wishes for accommodation for the clientDischarge accommodation decisions from rehabilitation (e.g., home, hostel, nursing home)Clinicians considered that “social situation, including personal and community supports” ranked 3rd most important cue, out of 15, in patient’s discharge accommodation decisions (mean rank 4.48 ± 6.03 variance); “Premorbid living arrangements” ranked 8th (7.90 + 9.99) and “Relatives’ choice/wishes for accommodation for the client” ranked 11th (9.71 ± 6.16).Unsworth 1995 [65]Baseline 1992; Quant; CS; Au, VIC; n = 74 consultants (in 13 teams)Stroke; clinicians in rehabilitation units68%F; age not collectedPremorbid living arrangements and Social situation (scaled from “no emotional or physical support of either a personal or community nature to assist the patient on discharge,” through to “constant support”) considered as 2 of 8 attributes pertinent when formulating a housing recommendationDischarge housing for 50 hypothetical patients from rehabilitation (Level 1: high dependency to Level 7: own home without supports or equipment)After personal mobility (beta: 0.465 + 0.087SD, weight range 0.266–0.611), social situation was 2nd (beta: 0.302 + 0.111SD, weight 0.173–0.548; with personal functional skills, beta: 0.308 + 0.111SD, weight 0.124–0.468) and premorbid living arrangements was 4th (beta: 0.164 + 0.066, weight 0072–0.294) in predicting ability to make an accommodation recommendation. The contribution of social support to the decision varied between beta 0.548 and 0.148 between the 13 teams, and premorbid living varied between 0.294 and 0.069. Unsworth 1996 [63]Baseline 1992; Quant; CS; Au, VIC; n = 62 (patients and consultants)Stroke; new cerebrovascular, inpatient rehabilitation > 6 d, age ≥ 60 y; and a patient’s rehabilitation team memberPatients (n = 58): 52%F; age 75 y (60–90). Team (n = 58): Demographics NRMarital status (Married, Single)Discharge decisions from rehabilitation (Level 1: total assistance to Level 7: completely independent)The team recommended and patients chose lower assisted housing if married (team: Level 5, patients Level 6; vs. single patients team: Level 3/4, t(58) 3.02, *p* = 0.0038; patients Level 5, t(49) 2.05, *p* = 0.0458). Patients saw discharge housing decisions as their own, unaware of rehabilitation team influence (correlation team recommendation vs. actual = 0.70) who generally recommended more support required. Unsworth 2001 [46]Baseline NR; Quant; CS; Au, VIC; n = 223Stroke (primary diagnosis); admitted using ICD-10 (World Health Organization, 1992) codes from 430 to 43853%F; age 77.14 y (60–93) Social interaction (measured in Adult FIM^SM^). Social situation, including personal and community supports. Of 24 variables assessedHospital discharge location (home, rehabilitation, nursing home)Social interaction, social support and premorbid housing were 3 of 6 variables (of 24 assessed) which predicted of discharge location, after additional adjustment. Social interaction predicted discharge to rehabilitation (model 1 coefficient = 2.884, model 2 = 1.741), home (2.411, 1.368), then nursing home (1.310, 1.260). Social situation predicted discharge to home (model 2 = 0.739), rehabilitation (0.538), then nursing home (−0.721). Premorbid housing predicted discharge to home (model 2 = 1.981), rehabilitation (1.884), then home (1.016).Unsworth 2003 [45]Baseline NR (5m); Quant; CS; Au, VIC, Melbourne; n = 60First stroke (primary diagnosis); hospitalised, aged ≥ 60 y, rehabilitation > 6 d52%F; age 74.7 y (60–90) Marital status. Premorbid living arrangements. Social situation. The latter two assessed of 8 cues.Discharge decisions from rehabilitation (Level 1: total assistance to Level 7: completely independent)Social support was the 3rd (b = 0.201) and premorbid housing was the 4th (b = 0.136) strongest predictor of higher independent housing recommendation (behind mobility 0.299, ADL 0.248). Single participants more likely to be discharged to supported housing (levels 1–3) than married participants (t(58) = 3.018, *p* = 0.0038). OUTPATIENT REHABILITATION ATTENDANCE

Fernandez 2008 [64]Baseline Aug–Nov 2005; Qual; CS; Au, NSW & ACT; n = 20 consultantsCoronary heart disease; cardiac rehabilitation coordinatorsDemographics NRQual (theme: Other individual barriers, subtheme: Low social support)Cardiac rehabilitation attendance “Lack of Quality Social Support: An effective system of social support is vital for the adoption of healthy behaviours. There is also evidence that involvement of partners in the rehabilitation process by engaging in risk reduction behaviours themselves is a critical factor in its effectiveness. Involvement of family in CR programs was discussed by most CR coordinators; however, they expressed that often, the family members did not actively engage in the risk modification behaviours. For example, a few coordinators indicated how the partner would smoke outside while their relative would attend CR, which demonstrated the lack of family support in engaging in risk modification behaviours.”Fernandez 2008 [60]Baseline Dec 2004–Mar 2005; Quant; CS; Au, NSW; n = 202CVD with successful percutaneous coronary intervention (PCI); survived 1–2 y, aged 18–80 y; not cognitively impaired; telephone contact number; hospital stay < 30 d post-PCI. Excluded: significant comorbidities; malignant disease; condition impairing cooperation in the study; transferred to a nursing home post-PCI27%F; age 64.0 ± 11.7 y (18–80)Marital status (living with partner vs. alone)Cardiac rehabilitation participationLiving with a partner was the factor most strongly associated with self-reported cardiac rehabilitation participation (OR 4.05; 95%CI 1.34–12.25; *p* = 0.013)Hagan 2007 [62]Baseline June–Aug 2000; Qual; CS; Au, VIC, Melbourne; n = 10Acute myocardial infarction; survivor, first-onset, sufficient English, referred to phase 2 cardiac rehabilitation20%F; age (31 + y)Living arrangements (alone vs. with another person); Qual (themes: Family support, Presence of support social networks)Cardiac rehabilitation attendance (perceived as a relevant goal)While not statistically significant, of the patients who attended rehabilitation, all lived with another person (n = 4 of 8) compared to none who lived alone (n = 0 of 2; *p* = 0.3, calculated from Table 1). (Qual) Family support: “Participants in this study frequently cited the importance of family support in providing meaning in their lives and the motivation to recover and make the necessary lifestyle changes… Although consideration of family members was often part of the participant’s decision to attend a phase 2 cardiac rehabilitation program, it was also discovered that in some instances family members had little influence over their actual decision to attend. Even though these participants stated that their families were supportive of their decision, they believed that this was coincidental to their decision to attend…However, issues or problems associated with the absence of family support were seen to impact significantly on the lives of some participants. Two participants lived alone at the time of the interview, and both discussed their lack of family support at some length…”Presence of support social networks: “If a person thought that his or her life was important to others, cardiac rehabilitation was likely to be seen as a relevant goal…Living alone raised unique issues related to the need for greater social support for some participants…In contrast with the participants who lived alone and failed to attend their scheduled appointments, this participant considered that attending a phase 2 cardiac rehabilitation program was a means of gaining social support networks.”Horwood 2015 [58]Baseline NR; Quant; CS; NZ, Dunedin; n = 44Coronary artery disease; event >6 m and completed outpatient rehabilitation >6 m, aged > 60 y29.5%F; age 72.7 + 6.9 y (60+)Partner status (married/living with partner vs. not). Perceived benefits (including Social Benefits: Social standing, At ease with people, Acceptance by others, Companionship) assessed using a Multi-dimensional Outcomes Expectations for Exercise Scale (MOEES). Perceived barriers (including Social Influences: Others with heart problems don’t go, Work responsibilities, Family responsibilities) measured using Cardiac Rehabilitation Barriers Scale (CRBS) Community-based cardiac rehabilitation attendanceNo association was observed between partner status and attendance (72.7% of high attenders were married/living with partner, 75% low attenders, 70.6% non-attenders, *p* = 0.648). High attenders at cardiac rehabilitation reported greater perceived social benefits in social standing (high 4.3 + 0.8, low 3.2 + 1.2, non 3.2 + 1.0, *p* = 0.022) and at ease with people (4.3 + 0.7, 3.4 + 1.0, 3.5 + 0.9, *p* = 0.036) than both low and non-attenders. No association was observed between attendance and acceptance by others (4.0 + 0.6, 3.7 + 1.0, 3.3 + 0.9, *p* = 0.129) or companionship (4.4 + 0.7, 3.8 + 0.9, 3.7 + 1.1, *p* = 0.151) or social influences (Others with heart problems don’t go 1.8 + 1.4, 1.4 + 0.7, 1.9 + 1.1, *p* = 0.446; Work responsibilities 1.6 + 1.2, 2.3 + 1.7, 2.0 + 1.4, *p* = 0.567; Family responsibilities 2.0 + 1.3, 1.9 + 1.2, 1.8 + 1.3, *p* = 0.937).Schulz 2000 [59]Baseline Jul 1993–Dec 1996; Quant; CS; Au, VIC, Horsham; n = 79Acute myocardial infarction; survivor, rural34%F; age NRMarital status (married, all other categories). Living with a partner (yes, no)Cardiac rehabilitation attendance Cardiac rehabilitation attendees were more likely to be married (vs. other categories; χ2 = 8.15, *p* = 0.004) and living with a partner (vs. not; χ2 = 7.58, *p* = 0.006) than respondents who did not attend.Sundararajan 2004 [38]Baseline Jan–Dec 1998; Quant; L (not defined); Au, VIC; n = 12,821Acute myocardial infarction (primary diagnosis); hospitalised, coronary artery bypass grafting or percutaneous transluminal coronary angioplasty as stated in the Victorian Admitted Episodes Dataset; survived ≥30 d based on the Victorian Deaths Registry29.9%F; age (40+) yMarital status (Currently married, Never married, Previously married, Unknown)The Victorian Cardiac Rehabilitation Dataset data linkage, attending ≥1 cardiac rehabilitation sessionBeing currently married was associated with attending cardiac rehabilitation (compared to: Never married OR 0.77, 95%CI 0.63–0.93; Previously married OR 0.77, 95%CI 0.68–0.87; Unknown OR 0.52, 95%CI 0.38–0.69)Thornhill 1998 [39]Baseline NR (recruited over 12m); Qual; CS; Au, NSW, Dubbo; n = 16Cardiac; hospitalised for a life-threatening cardiac episode, rural hospital50%F; age 60 y (47–53)Qual (theme: Important people, with subthemes: spouses, staff)Attending cardiac rehabilitation“Spouses were extremely important for people in both groups” (attending or non-attending cardiac rehabilitation). Partners participated in exercises, diet changes, attended rehabilitation (for attenders), and were great motivators. “Both groups of interviewees commented on two types of staff involved in their recruitment, or attempted recruitment… Interviewees, especially members of the attenders group, felt they had understood and remembered what helpful staff members said to them. Helpful staff were remembered because they chose times to see the interviewees when they ‘had some hope of understanding’ what was being said to them. Unhelpful staff were those who intimidated and confused participants.”REHOSPITALISATION



Korda 2017 [48]Baseline Jan 2006–Apr 2009 (to Dec 2011); Quant; L (30d); Au, NSW; n = 5074Heart failure; hospital diagnosis (primary or additional), participating in 45 and Up Study, age ≥ 45 y, with linked data, no death before discharge, had ≥30 d of follow-up, first readmission to hospital within 30 d follow-up was not planned42%F; age 80 ± 9.4 y, median 82 IQR 12 (45±)Marital status (single, de facto/married). Duke Social Support Index (DSSI) 4-item social interaction subscale, range 4–12.30 d unplanned readmission No association observed between marital status and 30 d unplanned readmission (aOR 0.91, 95%CI 0.78–1.05). Higher social interaction associated with lower 30 d unplanned readmission (aOR 0.95, 95%CI 0.91–0.99). Turner, 2010 [50]Baseline July 2003 to Jan 2006; Quant; L (6 w, 2.6 y); Au, NSW, Newcastle; n = 293–322 depending on missing dataCardiac; rehabilitation outpatients27.5%F; age 64.19 + 10.91 yMarital status (n = 293; married, single, widowed, divorced or separated). Living alone (n = 322; yes or no)Number of hospital admissions, mean ~2, follow-up 2.6 + 0.9SD yNo association observed between marital status or living alone with number of hospital admissions over 2.6 + 0.9SD y (univariable, statistics not reported). Winefield 1982 [47]Baseline 1980–1981; Quant; L (6–7 m); Au, SA, Adelaide; n = 29First myocardial infarction; hospitalised, men, age 30–65 y, English-speaking, completed psychological tests, survived 6–7 m. 0%F; age total sample: 53.53 + 8.18 y (30–65)The Social Environment Questionnaire (SEQ) question regarding confiders in respondent.Death, rehospitalisationMI patients who died or were hospitalised for cardiac treatment within 6 m reported fewer confidants (n = 8, 2.88 + 2.80SD; vs. not rehospitalised and survived n = 21, confiders: 4.81 + 2.93; *p* = 0.06).SURVIVAL 


Anderson 2004 [37]Baseline Mar 1981–Feb 1982; Quant; L (21y); NZ, Auckland; n = 680Acute stroke; participating in the first Auckland Region Coronary Or Stroke (ARCOS) Study51%F; age 71.2 ± 13.4 yEver married (not defined)Survival (not defined)No association observed between ever married and survival (92% deceased, 89% survived, X2 *p* = 0.37)Korda 2017 [48]Baseline Jan 2006–Apr 2009 (to Dec 2011); Quant; L (30d); Au, NSW; n = 5074Heart failure; hospital diagnosis (primary or additional), participating in 45 and Up Study, age ≥ 45 y, with linked data, no death before discharge, had ≥30 d of follow-up, first readmission to hospital within 30 d follow-up was not planned. 42%F; age 80 ± 9.4 y, median 82 y IQR 12 (45±)Marital status (single, de facto/married). Duke Social Support Index (DSSI) 4-item social interaction subscale, range 4–12.30 d unplanned readmission No association observed between marital status (aOR 1.06, 95%CI 0.77–1.47) or social interaction (aOR 0.97, 95%CI 0.88–1.07) and 30 d mortality.Stewart 2003 [61] Baseline Jun 1990–Dec 1992; Quant; L (baseline, median 8.1y); Au and NZ; n = 1130Acute myocardial infarction or hospitalized for unstable angina; survivors 3 m–3 y, age 31–75 y, enrolled in the LIPID study, a randomized placebo-controlled clinical trial of cholesterol-lowering treatment with pravastatin, and the LIPID Psychological Well-Being SubstudyGender NR; age (31–75 y) Divorce or separation (life events scale; yes in preceding year vs. not). Social isolation (living alone and ≤4/m vs. not). Marital/family problems (life events scale; yes in preceding year vs. not). General Health Questionnaire subscale of social dysfunction.Cardiovascular death (not defined), median follow-up 8.1 y No association was observed between divorce/separation (aHR 0.91, 95%CI 0.28–2.89), social isolation (aHR 0.69, 95%CI 0.43–1.10), social dysfunction (aHR 0.95, 95%CI 0.59–1.54), or marital/family problems (aHR 1.02, 95%CI 0.60–1.82) and cardiovascular death over mean 8.1 y.Turner, 2010 [50]Baseline July 2003 to Jan 2006; Quant; L (6w, 2.6y); Au, NSW, Newcastle; n = 293–322 depending on missing dataCardiac; rehabilitation outpatients27.5%F; age 64.19 + 10.91 yMarital status (n = 293; married, single, widowed, divorced or separated). Living alone (n = 322; yes or no)Mortality, follow-up 2.6 + 0.9SD y (39 d–3.8 y)No association observed between marital status or living alone with survival over 2.6 + 0.9SD y (univariable, statistics not reported). Wheeler 2012 [56]Baseline 2000–2002; Quant; L (5y); Au, SA; n = 337Acute myocardial infarction; hospitalised, participating in the Identifying Depression as a Comorbid Condition (IDACC) study which recruited cardiac and followed for 12 m post-discharge26%F; age 59 + 12 y for survivors; 69 + 11 y for fatalities (23–84)Perceived Social Support Scale (PSSS) All-cause mortality; Cardiac mortality, 5 yLower social support was a predictor of 5 y all-cause mortality (survivor 5.8 + 1.2SD, death 5.4 + 1.0, *p* = 0.08; aHR 0.70, 95%CI 0.54–0.90 *p* = 0.006) and cardiac mortality (survivor 5.8 + 1.2, death 5.3 + 1.2, *p* = 0.12; HR 0.67,95%CI 0.49 to 0.93, *p* = 0.016) after adjustment. Additionally adjusting for binary depression did not alter findings (all-cause HR 0.68, 95% CI 0.50 to 0.94, *p* = 0.019; cardiac NR).Wong 2010 [57]Baseline July 1994–June 2004; Quant; L (10y); Au, SA; n = 753Chronic heart failure; hospitalised, who had echocardiograms44.5% F; age 75.5 y Marital status (married, widowed, separated/divorced, single). Family support/Next of kin (child or spouse vs. other)Death linked the National Death Index, study census 30 June 2005“Other” relative (not spouse or child) as next of kin associated with increased survival (aHR 1.502, 95%CI 1.04–2.16, *p* = 0.028). Marital status not reported.Winefield 1982 [47]Baseline 1980–1981; Quant; L (6–7m); Au, SA, Adelaide; n = 29First myocardial infarction; hospitalised, men, age 30–65 y, English-speaking, completed psychological tests, survived 6–7 m. 0%F; age total sample: 53.53 + 8.18 y (30–65)The Social Environment Questionnaire (SEQ) question regarding confiders in respondent.Death, rehospitalisationMI patients who died or were rehospitalised for cardiac treatment within 6 months reported fewer confidants (n = 8, 2.88 + 2.80SD; vs. not rehospitalised and survived n = 21, confiders: 4.81 + 2.93; *p* = 0.06).* Social health includes marital/partner status and living alone/cohabiting as potential proxies. Mean + SD age reported unless otherwise indicated. Acronyms: 95%CI—95% confidence interval; aHR—adjusted hazard ratio; aOR—adjusted odds ratio; Au—Australia; CS—cross-sectional; d—days; F—female; HR—hazard ratio; IQR—interquartile range; L—longitudinal; m—months; NR—not reported; NSW—New South Wales; NZ—New Zealand; OR—odds ratio; *p*—*p*-value; QLD—Queensland; Qual—qualitative methods; Quant—quantitative methods; RR—relative risk; SA—South Australia; SD—standard deviation; TAS—Tasmania; uOR—unadjusted odds ratio; VIC—Victoria; vs.—versus; WA—Western Australia; w—weeks; y—years.

### 3.3. Outcomes

We grouped the included studies based on outcomes, creating five themes, presented in Table 1 by anticipated sequential order from CVD event to death: inpatient length of stay, discharge destination, outpatient rehabilitation, rehospitalisation, and survival. To be clear on the definition, outpatient rehabilitation refers to when a patient resides at home and receives recovery or rehabilitation therapy at a hospital, clinic or in their home. Outpatient rehabilitation often involves an individualized treatment plan through a complete assessment undertaken during the first visit. The therapy can be delivered individually or in groups to assist reaching individual goals. Four studies reported multiple relevant outcomes and are represented multiple times [47,48,50,63].

#### 3.3.1. Inpatient Length of Stay

Two quantitative Australian studies assessed three social health measures as possible contributors to the length of inpatient stay. Both reported that marital status was not associated with the length of stay. No studies assessed social isolation, social support or loneliness as possible contributors to the length of stay. Among 58 stroke patients, the length of rehabilitation stay was not associated with marital status [63]. Among ~322 cardiac rehabilitation outpatients, hospital length of stay was not associated with living circumstances (living alone vs. not) or being married (compared to being single, widowed, divorced or separated) after adjustment for disease severity, depression and anxiety [50].

#### 3.3.2. Discharge Destination

Ten Australian cross-sectional studies, predominately involving stroke patients, assessed 22 social health measures as possible contributors to discharge destination. Overall, each social health measure was important for the discharge destination, with positive social health being associated with a higher level of independent living, including admission to rehabilitation rather than a nursing home. Many studies assessed living situation (n = 7) and social support (n = 8).

Premorbid living situation was ranked as important for discharge decision from rehabilitation by 74 consultants (4th of 8 attributes; on a graded dependency scale) [65] and 85 clinicians (8th of 15 cues; e.g., home, hostel nursing home) [40]. To be clear on the definition, a “hostel” refers to an aged care service in which residents receive personal care and accommodation support [67]. The majority of hostels also provide some nursing care. Studies unanimously reported that premorbid living at home with others was associated with admission to rehabilitation or being discharged to a higher level of independent housing; among 14 consultants assessing 75 patients [52] for rehabilitation, it was premorbid living with others, rather than home alone or supported accommodation; among 223 rehabilitation stroke patients [46], it was to home or rehabilitation, rather than a nursing home; among 60 stroke patients [45], premorbid housing was the 4th (of 8) strongest predictor of higher independent living (on seven levels graded from low to high independence); and among 616 stroke patients [51], premorbid living at home with others was associated with discharge to rehabilitation rather than a nursing home (compared to patients who lived in supported accommodation/hostel).

Two studies among 64 patients and consultants [63] and 60 stroke patients [45] reported that married participants were more likely to be discharged to higher independent housing (on seven levels graded from low to high independence). Generally, the presence of a spouse/carer/relative was associated with being admitted to rehabilitation,; however, the ability to provide support and accept the goals of rehabilitation were also important among 14 consultants assessing 75 patients [52] and 21 consultants [54,55]. Interpersonal relationships were also discussed by Aboriginal patients who reported ”competing family and community obligations” and ”loneliness and dislocation from family and community” as reasons for high hospital self-discharge rates [49].

Social isolation (measured as low levels of social interaction) was the second strongest predictor (of 24 variables) of discharge destination to a nursing home, then home, rather than rehabilitation, among 223 stroke patients [46].

Two qualitative studies of 14 consultants assessing 75 patients [52] and 17 physicians [53] reported that positive social factors were one of the most influential or important factors affecting discharge to rehabilitation (versus not), with the former study [52] reporting 4.4-fold increased odds. Quantitative studies also reported that social support was ranked second (of 8 attributes) by 74 consultants [65] and third (of 15 cues) by 82 clinicians [40] as most important for discharge decisions from rehabilitation. Social support was associated with recommendations for higher levels of independent housing among 60 stroke patients (ranked third of eight) [45] and among 223 stroke patients (to home, then rehabilitation, rather than a nursing home) [46].

#### 3.3.3. Outpatient Rehabilitation Attendance

Seven, predominately Australian, studies assessed nine social health factors as potential contributors to outpatient rehabilitation aspects, with three using qualitative methods. Overall, living situation and social support measures were associated with higher attendance at outpatient rehabilitation; however, findings were conflicting for partner status.

Generally, living with a spouse or others was associated with greater attendance at outpatient rehabilitation, but there were conflicting findings regarding simply having a spouse. Three quantitative analyses among 79 myocardial infarction participants [59], 10 myocardial infarction participants [62] and 202 percutaneous coronary intervention participants [60] reported that living with a partner or others was associated with attending cardiac rehabilitation, with the latter [60] reporting it was the most strongly associated factor (compared to age, gender and income).

Similarly, two quantitative studies among 79 myocardial infarction participants [59] and 12,821 patients [38] reported that being married was associated with attending cardiac rehabilitation. However, one quantitative study among 44 coronary artery disease New Zealanders [58] reported no association, and a qualitative study among 16 cardiac participants [39] reported that “Spouses were extremely important for people in both groups” (attending or non-attending).

Generally, social support, from staff or family, was associated with attendance at outpatient rehabilitation; however, one study [64] made the observation that, to be effective, the support person/s needed to engage in risk-reduction behaviours themselves. Among 44 coronary artery disease patients in New Zealand [58], quantitative analysis revealed that high attenders were more likely to report social benefits, including social standing and being at ease with people, but were no more likely to report other social benefits, such as acceptance by others and companionship, or barriers from social influences (such as others with heart problems do not go, work responsibilities, and family responsibilities) than low attenders. Among 16 cardiac participants [39], qualitative synthesis revealed that socially supportive staff, who “chose times to see the interviewees when they ‘had some hope of understanding’ what was being said to them” rather than staff “who intimidated and confused participants”, was associated with rehabilitation attendance. Among 10 myocardial infarction participants [62], qualitative synthesis revealed the themes of family support and the presence of social networks. Family support was frequently cited as “part of the participant’s decision to attend” and “providing meaning in their lives and the motivation to recover” [62]. Additionally “absence of family support was seen to [negatively] impact significantly on the lives of … Two participants [who] lived alone” [62]. Similarly, the presence of social networks assisted with the participant’s motivation to attend and “Living alone raised unique issues related to the need for greater social support” [62]. However, one “participant considered that attending a phase 2 cardiac rehabilitation program was a means of gaining social support networks” [62]. However, another qualitative study among 20 consultants of coronary heart disease patients [64] reflected that “involvement of partners in the rehabilitation process by engaging in risk reduction behaviors themselves is a critical factor in its effectiveness”. In discussing low-quality social support, the authors explain that “An effective system of social support is vital for the adoption of healthy behaviors… however, they expressed that often, the family members did not actively engage in the risk modification behaviors. For example…the partner would smoke outside while their relative would attend CR [Cardiac Rehabilitation]” [64].

#### 3.3.4. Rehospitalisation

Three quantitative, longitudinal Australian studies assessed five social health factors as potential contributors to rehospitalisation. Overall, proxies of social health (partner status or living situation) were not associated with rehospitalisation, but social isolation and social support measures were.

Partner status was not associated with rehospitalisation in two studies involving 293 and 5074 participants [48,50]. Living situation was not associated with rehospitalisation over 2.6 years in one study involving 322 cardiac rehabilitation outpatients [50]. Higher social interaction (i.e., lower levels of social isolation) was associated with lower 30-day unplanned readmission among 5074 patients hospitalised for heart failure [48]. Higher social support (SEQ) was associated with less rehospitalisation and better survival over six months among 29 Australians hospitalised for their first myocardial infarction [47].

#### 3.3.5. Survival

Seven quantitative longitudinal studies assessed ten social health factors as potential contributors to survival. In general, proxies of social health (partner status or living situation) and social isolation were not associated with survival, but social support measures were.

Partner status was not associated with survival in four studies involving 293, 680, 1130 or 5074 participants followed for up to 21 years [37,48,50,61]. Living situation was not associated with survival over 2.6 years in one study involving 322 Australian cardiac rehabilitation outpatients [50]. “Other” relative (not spouse or child) provided as next of kin was associated with increased survival in one study involving 753 Australians hospitalised for chronic heart failure [57].

Social isolation was not associated with 30-day survival among 5074 Australians [48] or with cardiovascular death over 8.1 years (mean) among 1130 Australians and New Zealanders [61]. Social support was associated with survival in two studies. Among 337 Australians hospitalised for myocardial infarction (mean age 59, 26% female), lower social support (PSSS) was a predictor of 5-year all-cause mortality and cardiac mortality in fully adjusted models [56]. Higher social support (SEQ) was also associated with less rehospitalisation and better survival over six months among 29 Australian men hospitalised for their first myocardial infarction [47].

## 4. Discussion

We grouped the 25 included papers (from 24 study cohorts) based on outcomes, creating five themes for health service utilisation and survival after a CVD event. Positive social health was consistently associated with the discharge destination theme, characterised by higher independent living (including admission to rehabilitation rather than a nursing home). Social support was the most frequently reported social health measure or qualitative theme (n = 17) and was consistently associated with better outcomes across four themes (discharge destination, outpatient rehabilitation, rehospitalisation and survival), but was not assessed as a predictor of length of stay (in hospital or rehabilitation). Several studies considered the social health proxies of partner status (n = 12) and living situation (n = 10); however, they were inconsistent regarding outpatient rehabilitation and were not associated with length of stay, rehospitalisation or survival. Very few studies considered social isolation (n = 4) or loneliness (n = 1) as possible contributors to health service utilisation and survival after a CVD event among Australians and New Zealanders.

Very little identified in this review is directly comparable to the prior systematic reviews [20,21,22], as the only overlapping themes are hospital length of stay [20] and hospital readmissions [20,21,22]. In our review, we did not identify any study that assessed social support and length of stay. Aligned with prior systematic reviews [20,21,22], we report a single study that observed that higher social support was associated with less rehospitalisation and better survival over six months among myocardial infarction patients [47]. These findings support the theory of ‘*social admission*’, which can be described as a patient with no acute medical needs but no available safe discharge option due to their social circumstance [23]. For example, informal caregivers may no longer be available or are unable to cope with the patient’s needs, or there is simply no one available for in-home support. ”Acopia” and ”bed-blocker” have also been used to describe this phenomenon [23].

More generally, we found that greater social support was consistently associated with better outcomes across discharge destination, outpatient rehabilitation, rehospitalisation, and survival themes. Patients with higher social support were more likely to be discharged to higher independent living options. If admitted to rehabilitation, it was usually in comparison to the alternative higher-cost-burden option of a nursing home. Once admitted to outpatient rehabilitation, greater social support was associated with greater attendance, representing a potentially greater cost burden to society. However, rehabilitation is known to improve health and lower subsequent CVD risk, which likely presents a cost-saving scenario in the long term. The benefits of rehabilitation attendance are likely reflected in our findings of higher social support being associated with lower rehospitalisation and greater survival among cardiac patients.

Four studies [49,52,54,55,64] noted that it was important for the social supports to be accepting of and role-model the CVD patient’s healthy behaviour rehabilitation goals. Two studies reported that although the presence of a spouse/carer/relative was beneficial, their ability to provide support and accept the patient’s goals were also important for the consultant’s or physician’s decision for rehabilitation admission [52,54,55]. Another study [64] articulated this point by stating that the supports needed to engage in the risk reduction behaviours themselves for outpatient rehabilitation to be effective. The rehabilitation consultants “expressed that often, the family members did not actively engage in the risk modification behaviors” and provided the example of the supports smoking outside while they wait for the patient to attend rehabilitation [64]. “*Competing family and community obligations*” were also discussed among Aboriginal patients as reasons for high hospital self-discharge rates [49]. These examples of negative support align with prior literature, demonstrating supports may be unhelpful or constraints during healthcare delivery [11].

Our systematic review demonstrated that social health was consistently associated with discharge destination decisions. Being married, living at home with others, the presence of a spouse/carer/relative, not being socially isolated or lonely, and having social support were associated with being discharged to rehabilitation or higher independent living. Discharge destination was the only theme where social health proxies (partner status and living status) aligned with social isolation, social support and loneliness findings. Our systematic review suggests that partner status and living status should not be considered social health proxies of health service utilisation and survival after a CVD event. Again, comparability to the 2018 systematic review [20] is not possible, as this study only assessed social support.

Our systematic review assessed social health as the exposure after a CVD event. We acknowledge that CVD may impair, benefit or not alter a patient’s social health. For example, among 48 community-dwelling adults who had undergone coronary artery bypass graft surgery (mean age: 66.6 ± 9.9SD, 14.6% female), social functioning was reduced two years after surgery [68]. In a qualitative study [69] of 12 adults at 1, 3, or 5 years post-stroke discharge (mean age 72 + 14SD years, 50% female), there was evidence of lower socialising and the effects of social isolation, described as “*I miss them*”. Most survivors reported being “*confined*” and that it was difficult to attend events outside the house due to reduced physical ability, mobility restriction and transport issues [69]. A few participants started new activities, but these were stroke or hostel related [69]. Among 143 patients aged under 65 with their first myocardial infarction, the belief that their heart disease would have serious consequences was associated with reduced recreational activities and social interaction [66]. However, no change is also possible, as observed among 13 stroke survivors, where social relationships did not change between 3 weeks and 3 months post-discharge and were not different to the general population [70]. Hence, social health changes associated with a CVD event may not influence inpatient length of stay or discharge destination but may influence longer-term outcomes. Alternatively, health service utilisation after a CVD event may provide opportunities to gain social support. Within our systematic review, we noted that two studies [58,62] reported that participating in cardiac rehabilitation could increase social support. Additionally, there is evidence that education and counselling interventions for families can maintain family functioning and, in turn, lead to improved functional and social patient outcomes [71].

### 4.1. Strengths and Limitations

A strength of our review is that we included all settings across the healthcare system. An alternative approach for future research could be to conduct a systematic review for each theme, incorporating more countries. Notably, prior apprehensions regarding the potential null association between social health and cardiovascular disease in Australia [13] have been overcome with a validated, medically diagnosed measure of CVD (rather than prior publications assessing self-report CVD) [6]. While each country has unique social and healthcare systems and population characteristics, there are no specific reasons why this review is not generalisable to other high-income countries.

As acknowledged in the sibling review [12], we may have overlooked publications where the authors did not describe partner status or living situation as social health constructs or that were published after June 2020. We acknowledge that medical knowledge, technology and treatment has improved greatly over the included article publication timeframe (from 1982). Overall, we observed a similarity in findings despite the date the research was undertaken, adding weight to the strength of our findings that social support and social isolation are associated with health service utilisation and survival after a cardiovascular disease event. Future research could evaluate whether social health influences care decisions and health outcomes for other chronic diseases, and among other countries with different social or health care systems.

Assessment of the demographics of the included studies revealed that women were under-represented across the studies included in this review (8 of 20 studies reported <40% female). CVD carries a high prevalence and burden amongst women (responsible for nearly one-third of all deaths among women), and there has been increasing recognition that aspects of CVD prevention and treatment are unique to women [72]. Further, gender may play a role in cardiac recovery. Women are less likely to complete cardiac rehabilitation after acute coronary syndrome and may require more social support to encourage attendance [73]. For example, having social support may be limiting if the woman is the primary caregiver in the family, as this would reduce their availability for cardiac rehabilitation [73]. However, women may be more encouraged by positive social supports from family and peers, which has been shown to improve their coping mechanisms for attending the rehabilitation program [73]. Our study adds to the importance of research focused on women and the importance of gender-disaggregated reporting.

### 4.2. Clinical Significance

Healthcare professionals are placed in a position to help with barriers that affect a person’s emotional, social, and physical needs. Our findings highlight that social health plays a role in discharge destination and, therefore, where and how healthcare is being delivered (i.e., outpatient, rehabilitation, or nursing home). Furthermore, healthcare professionals can also play a large role in the decision of where patients will be discharged [40,53,54,55,63,65] and may be unconsciously or informally incorporating social health into their decision. Acknowledging that social health is part of the decision-making process is the first step to incorporating social health into formal procedures. Additionally, healthcare professionals should be aware that social support is a factor that needs to be considered during recovery to achieve the best outcomes in outpatient rehabilitation attendance, rehospitalisation and survival. Incorporating a formal assessment of social support into healthcare management plans will likely improve outcomes and survival after a cardiovascular disease event. This formal assessment of social support needs to account for negative support, such as willingness and unhealthy lifestyle behaviours, which could impede a patient’s recovery.

Improving someone’s social health through cardiac recovery could be an avenue to benefit economic costs to both the patient and the healthcare system. Assessment as to how healthcare workers can approach and intervene to improve social health is required. Future research should identify how patient social health needs can be improved through clinical and community settings, healthcare systems, families, and social services. Addressing negative supports may also be beneficial to the support and the patient’s cardiac recovery. A systematic review identified early evidence for enhancing social support through caregiver-oriented strategies for people living with CVD [74]. In comparison, the systematic review reported inadequate evidence to promote cognitive behavioural therapy, mindfulness, peer support, and multi-faceted cardiac rehabilitation programmes for increasing social support for people living with CVD [74]. Incorporating supports in the patient’s cardiac rehabilitation program may also be a solution to reducing the negative consequences of becoming a caregiver, such as caregiver strain and increased social isolation. A scoping review identified that informal caregivers to people who had just had a stroke “lost their social and leisure activities, which made them feel unhappy and socially isolated” [75]. While caregivers increased their social activity levels over time, some remained without social and leisure activities, which, in turn, increases the risk of depression [75]. The review concluded that there was a lack of interventions to assist caregivers in maintaining their social and leisure activities [75].

## 5. Conclusions

Our review demonstrates that social support is considered in cardiac care decisions and adds to the evidence more broadly that lower social support is associated with higher-intensity healthcare services and poorer survival. Further assessment is required to evaluate the impact of social isolation and loneliness on health service utilisation and survival after a CVD event. As partner status and living status did not align with social isolation and social support findings in this review, we recommend they not be used as social health proxies when assessing health service utilisation and survival among CVD patients.

Overall, among Australian and New Zealand cardiac patients, positive social health was consistently associated with discharge to higher independent living, including admission to rehabilitation rather than a nursing home. Greater social support was consistently associated with better outcomes for discharge destination, outpatient rehabilitation, rehospitalisation and survival. Several studies reflected that it was also important for the social support to be accepting and role-model the CVD patient’s healthy behaviour goals. Therefore, healthcare professionals need to be aware that social support is a factor to be considered to achieve the best patient outcomes, with the potential of negative and positive supports.

Further assessment as to how healthcare workers can approach and intervene to improve social health is required. Future research should identify how patient social health needs can be improved through intervention, clinical and community settings, healthcare systems, families, and social services. Incorporating the people who are providing social support, particularly those who display unhealthy behaviours, into the patient’s CVD rehabilitation program could have benefits for the supportive person and patient outcomes. Lastly, our review highlights the need for future CVD research to focus on women and the importance of gender-disaggregated reporting.

## Figures and Tables

**Figure 1 ijerph-20-04853-f001:**
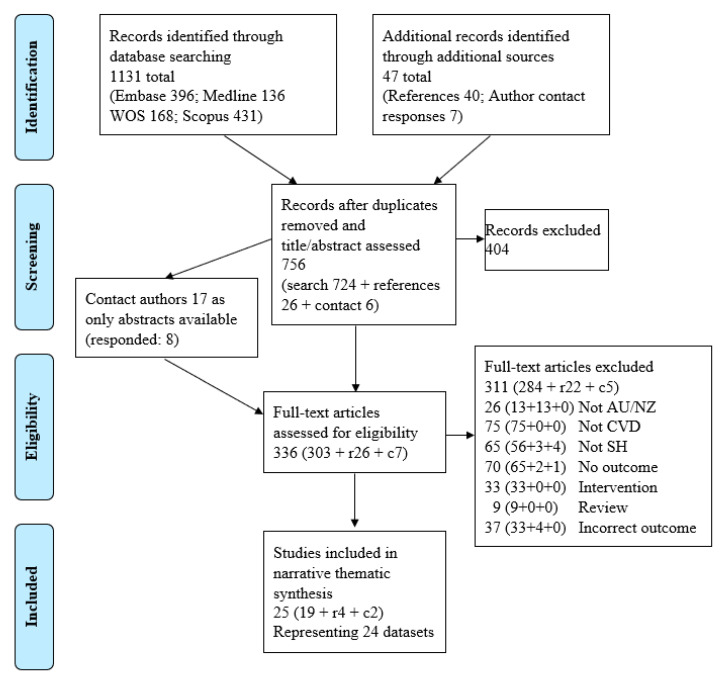
Flow Diagram of Review Process. Acronyms: AU—Australia; c—contact (number of papers from contact with authors of included papers); CVD—Cardiovascular Disease; Datasets—number of study cohorts; NZ—New Zealand; r—references (number from searching the references of included papers); SH—Social Health; Studies—individual papers; WOS—Web of Science.

## Data Availability

Not applicable.

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
