# Peer review of "Social Isolation and Social Support Influence Health Service Utilisation and Survival after a Cardiovascular Disease Event: A Systematic Review"

_ijerph, 2023, doi:10.3390/ijerph20064853_

Round 1
Reviewer 1 Report
I congratulate the authors for the paper and the theme.
The paper describes how social isolation, social support, or loneliness influence health service utilization and survival after a cardiovascular disease event.
Title: Methodologically, the title describes it as a narrative thematic systematic review. Doesn't it explain in the method section what a narrative thematic systematic review is? What is your theoretical framework?
This concept - narrative thematic is used 5 times throughout the paper: 1st in the title; 2nd in the summary; 3rd in a subtitle; and the remaining two in the title of two bibliographic references, both by… Freak-Poli et al ( 2020/ 2021)
In systematic reviews, is thematic analysis the data analysis strategy?
Maybe reconsider the title...
I also warn that the titles of Freak-Poli are very similar to the present article.
Freak-Poli R, Hu J, Phyo AZZ, Barker F. Does social isolation, social support or loneliness influence health or well-being after a cardiovascular disease event? A narrative thematic systematic review. Health & social care in the community. 2021.
Freak-Poli R, Hu J, Phyo AZZ, Barker F. Does social isolation, social support or loneliness influence health or well-being after a cardiovascular disease event? A narrative thematic systematic review. Health & social care in the community. 2022;30(1):e16- 566 e38
Introduction – it is organized and easy to read, framing the article properly. However, certain statements such as ( line 40) “driven largely by direct healthcare cost (55%)” – the suggestion is to include a reference.
In the sentence ( Line 57 to 59) “However, the review also observed associations between lower social support with increased rates of readmission to hospital, and smaller social networks with longer hospital stays” – suggestion: put reference.
Review the paper, some words are together (line 79): Hence, we restrictedthe search bas
Methodology. It is fragile and the results may be biased. First because:
Line 156 to 158 - “Initially, we aimed to assess the influence of social health on a CVD patient’s journey, including health service utilisation, health, quality of life and survival. However, the search provided an unmanageable number of included studies (>70) and outcomes of interest (>10).”
This constitutes a bias – the authors possibly did not accurately address the starting question/objective; it also does not assess the methodological quality of the included articles;
Line 105, mention that “ This systematic review and meta-analysis were conducted in accordance with the Preferred Reporting Items for Systematic Reviews and Meta-Analyses (PRISMA)”
In fact, this review is not rigorously a systematic review (the methodological quality assessment is lacking, for example; qualitative data and quantitative data are combined - for that they would have to follow the methodology of mixed studies), but it is accepted; however, it is not a meta-analysis at all. Therefore, the suggestion is to rephrase the sentence.
Line 124 - Another fragility is the moment of the search (2 years ago), but as the authors very well mentioned in the discussion, this constitutes a fragility. Therefore, understanding that it could happen and as the information is clear, it does not compromise future revisions on the subject.
Line 163 to 170 - I would like you to clarify this information – “Research manuscripts reporting large datasets that are deposited in a publicly available database should specify where the data have been deposited and provide the relevant accession numbers. If the accession numbers have not yet been obtained at the time of submission, please state that they will be provided during review. They must be provided prior to publication. Intervention studies involving animals or humans, and other studies that require ethical approval, must list the authority that provided approval and the corresponding ethical approval code.”
This is confusing information and I fear for other readers as well.
Results:
Line 191 to 194 – “Twenty-five papers, representing 24 studies, were included in the review: 19 from the original search, 4 from references and 2 from contact with authors (Table 1).” It is not clear: there are 25 papers, but one is not a study, so what is it?
The inclusion criteria refer to: Cross-sectional and longitudinal observational data (review articles were excluded).
It seems unclear.
With rigor when describing the information in the results, the references of the articles included must accompany the information described. For example:
Line 201 to 204 - CVD measure Recruitment was focused on stroke (n=10),( WHICH STUDIES?) myocardial infarction (n=5)( WHICH STUDIES?) ; one combined with angina, another combined with coronary artery bypass grafting ),( WHICH STUDIES?) and percutaneous transluminal coronary angioplasty), coronary disease (n=2) ),( WHICH STUDIES?) , heart failure (n=2) ),( WHICH STUDIES?) , percutaneous coronary intervention (n=1) ),( WHICH STUDIES?) or CVD more broadly (n=3) ),( WHICH STUDIES?)
Total: 10 + 5+ 1+ 2+2+1+3 = 24. Did you not include 25 studies?
Suggestion: Put legend in Prisma, data that uses several acronyms
Line 214 to 217: One study (WHICH?) used the social health scale Perceived Social Support Scale (PSSS). Four (WHICH?) studies used social health sub-scales from the Duke Social Support Index (DSSI), …it is important to put the references of the studies.
In the sentence - The majority of studies recruited participants from Australia (n=21) while only two recruited from New Zealand and one recruited from both countries
The same confusion continues with the 25 papers included, but which only describe 24
Table 1 is not a reader-friendly table. Too much information, structurally not pleasant. Column headings do not always translate the content: example 1: Title “ social Health” – then put data Marital status ; example 2: column title – Sample: they put a lot of information, in addition to the sample, such as methodology, country, city …
Suggestion: try to put the most user friendly table and review: for example -
Hayward 2014 [Full Text + Abstract] - take [Full Text + Abstract];
Fernandez 2008 JCN (60)- JCN ?
Discussion
Could you make this information clearer for the reader ( line 376 to 378)
Authors should discuss the results and how they can be interpreted from the perspective of previous studies and of the working hypotheses. The findings and their implications should be discussed in the broadest possible context. Future research directions may also be highlighted.
Line 414 to 415: Four studies (50, 52-54, 60) noted that it was important for the social supports to be accepting of and role-model the CVD patient’s healthy behavior rehabilitation goals
But they reference 5 = 50,52,53,54, 60…
The same problem (line 416 to 415): Two studies reported that although the presence of a spouse/carer/relative was beneficial, their ability to provide support and accept the patient’s goals were also important for the consultant’s or physician’s decision for rehabilitation admission (50, 52, 53)
Reviewer 2 Report
Dear authors, thank you for your interesting contribution. Here are some of my suggestions.
INTRODUCTION: as mentioned, social support is not always supportive. Please, clarify this point better by adding a social support definition and possible cognitive, emotional, and behavioral consequences (see Sebri et al., 2021). Moreover, define how social support could be helpful for cardiovascular patients (see Freak-Poli et al., 2021). I argue that each type of illness has different requests and needs
METHODS: considering only some countries could be a discussion point in order to have a larger overview of literature knowledge. Please, clarify this choice. Second, why did you not involve PSYCHINFO or other psychological sources specifically)? Why are exclusion criteria not presented? Please, add them. Third, have you previously registered this study in PROSPERO? Please, provide the number registration. Lastly, I am not sure that the lack of age restrictions would be an appropriate choice. Please, motivate your choice.
RESULTS: Please, do not report participants' citations in Table 1.
DISCUSSION: Add literature references to sustain your findings, please. Moreover, future research has to be explored.
REFERENCES
- Freak-Poli, R., Ryan, J., Neumann, J. T., Tonkin, A., Reid, C. M., Woods, R. L., ... & Owen, A. J. (2021). Social isolation, social support and loneliness as predictors of cardiovascular disease incidence and mortality. BMC geriatrics, 21(1), 1-14.
- Sebri, V., Mazzoni, D., Triberti, S., & Pravettoni, G. (2021). The impact of unsupportive social support on the injured self in breast cancer patients. Frontiers in Psychology, 12, 722211.
Round 2
Reviewer 1 Report
There is one aspect that I would like to say to the authors, who possibly got confused by my observation.
[Line 105, mention that " This systematic review and meta-analysis were conducted in accordance with the Preferred Reporting Items for Systematic Reviews and Meta-Analyses (PRISMA)"]
In your method, you should say something like:
The study selection was oriented according to the Preferred Reporting Items for Systematic Reviews and Meta-Analyzes Extension for Scoping Reviews (PRISMA) diagram.[ put reference]
The question is that you said that your meta-analysis... ( This ... meta-analysis..)
Then you should also add something like: Figure 1 shows the study selection process in the form of a PRISMA diagram [put reference]
Summary: saying that you follow PRISMA is important. Excuse my lack of objectivity in the comment made, which led you to remove something that was fine. What you need is to reformulate, ok?
Success votes for the publication
Author Response
Thank you for bringing our attention to this reference, which we had not yet come across. We have updated the methods:
... the registered protocol (Prospero CRD42020099557). The study selection was oriented according to the Preferred Reporting Items for Systematic Reviews and Meta-Analyzes Extension for Scoping Reviews (PRISMA) diagram [80]. The only other change ...
Please note that we have retained reference to Figure 1 for the results section.
kind regards, The Authors